# Urinary Volatile Organic Compound Analysis for the Diagnosis of Cancer: A Systematic Literature Review and Quality Assessment

**DOI:** 10.3390/metabo11010017

**Published:** 2020-12-29

**Authors:** Qing Wen, Piers Boshier, Antonis Myridakis, Ilaria Belluomo, George B. Hanna

**Affiliations:** Department of Surgery and Cancer, Imperial College London, St Mary’s Hospital, London W2 1NY, UK; q.wen17@imperial.ac.uk (Q.W.); piers.boshier03@imperial.ac.uk (P.B.); a.myridakis@imperial.ac.uk (A.M.); i.belluomo@imperial.ac.uk (I.B.)

**Keywords:** volatile organic compounds, urine, cancer early detection, mass spectrometry

## Abstract

The analysis of urinary volatile organic compounds (VOCs) is a promising field of research with the potential to discover new biomarkers for cancer early detection. This systematic review aims to summarise the published literature concerning cancer-associated urinary VOCs. A systematic online literature search was conducted to identify studies reporting urinary VOC biomarkers of cancers in accordance with the recommendations of the Cochrane Library and Meta-analysis of Observational Studies in Epidemiology (MOOSE) guidelines. Thirteen studies comprising 1266 participants in total were included in the review. Studies reported urinary VOC profiles of five cancer subtypes: prostate cancer, gastrointestinal cancer, leukaemia/lymphoma, lung cancer, and bladder cancer. Forty-eight urinary VOCs belonging to eleven chemical classes were identified with high diagnostic performance. VOC profiles were distinctive for each cancer type with limited cross-over. The metabolic analysis suggested distinctive phenotypes for prostate and gastrointestinal cancers. The heterogenicity of study design, methodological and reporting quality may have contributed to inconsistencies between studies. Urinary VOC analysis has shown promising performance for non-invasive diagnosis of cancer. However, limitations in study design have resulted in inconsistencies between studies. These limitations are summarised and discussed in order to support future studies.

## 1. Introduction

There remains an important unmet clinical need to improve the earlier detection of cancer. Early symptoms of many forms of cancer are often vague and may be mistaken for common benign conditions. Without access to acceptable and affordable methods of assessing patients who present with such symptoms, diagnosis is often delayed until the cancer is at an advanced, sometimes incurable, stage.

There is growing evidence linking different cancers to the increased/decreased production of volatile organic compounds (VOCs) [1,2,3,4,5,6]. As end products of metabolism, cancer-related VOCs are potentially produced by oxidative stress and peroxidation of cell membranes or as a consequence of gene or protein alterations in cancer cells. VOCs levels can reflect pathophysiological processes including inflammation, necrosis, and cancer development [7,8]. Owing to their volatility at ambient temperature VOCs produced within cancer tissues travel in the systemic circulation before being freely excreted. In humans, VOCs have been detected in a wide range of samples, including breath, urine, blood, faeces, tissue, and skin.

Compared to other biological samples such as breath, blood, and tissue, urine has the advantage of being easy and inexpensive to collect and handle. Furthermore, urine has the potential to provide insight not only to local environment of the urogenital tract but also systemic metabolism as it contains the effluent of renal filtration. Studies that have analysed urinary VOCs in cancer patients have yielded promising results [9,10,11]. Hanai et al. reported nine VOCs that were present at higher concentrations in urine of lung cancer patients [12]. Arasaradnam et al. demonstrated for the first time the utility of urine specimens to discriminate healthy individuals from pancreatic ductal adenocarcinoma (PDAC) patients through the detection of VOCs. Their findings were also able to distinguish between early and late stage PDAC [10].

Despite promising findings, there remains uncertainty as to the role of urinary VOC analysis in cancer early detection. A major perceived challenge is the lack of standardisation of methods that may hinder efforts to achieve reproducibility of findings and in turn wider adoption in clinical practice.

The purpose of this systematic review is to summarise the published literature concerning cancer-associated urinary VOCs. Specific objectives were to identify published urinary VOC markers of cancer; explore emerging metabolic pathways of cancer specific VOCs; and evaluate the methodological quality of published studies.

## 2. Results

The systematic literature search identified 886 studies. After screening and assessment for eligibility, a total of 13 studies were included (Figure 1) [9,11,13,14,15,16,17,18,19,20,21,22,23]. Details of included studies are presented in Table 1. Included studies were from Europe (n = 10), North America (n = 2), and Asia (n = 1) reporting the outcomes of 1266 patients of which 700 had been diagnosed with cancer. Studies reported the urinary VOC profiles of prostate cancer (n = 5); gastrointestinal cancer (n = 7); lung cancer (n = 1); haematological malignancies (n = 1); and bladder cancer (n = 1).

Gas chromatography mass spectrometry (GC-MS) was the most commonly used analytical technique (10 studies). Selected ion flow tube mass spectrometry (SIFT-MS) was used in three studies. Field asymmetric ion mobility spectrometry (FAIMS) was used in a single study. Two studies used more than one analytical technique [14,21]. In the majority of studies (n = 9), VOCs were analysed within urine headspace, as opposed to the fluid phase. Other techniques used for the extraction of urinary VOCs included solid phase microextraction (SPME) and stir bar sorptive extraction with or without derivatisation.

### 2.1. Quality Assessment

Outcomes of quality assessments are summarised in Table 2. Bias and applicability of outcomes were analysed with QUADAS-2 (Table 2). Of the 13 included studies, there was considered to be an overall low risk of bias and high applicability of these studies to the review question.

General reporting quality of the studies was assessed by the STARD checklist (Table 2). STARD scores ranged from 11 to 27 with a mean of 19.9 ± 4.6 where the maximum score is 41, indicating that reporting standards were often inadequate.

Reporting of metadata in metabolomics datasets was assessed using CAWG-MSI (Appendix A) [24]. Only three studies reported greater than 50% of the CAWG-MSI criteria [13,14,18]. Eight of the 13 included studies used a relative quantification of compounds [11,13,14,15,16,20,21,23], whilst five studies provided an absolute quantification of compounds [9,17,18,19,22]. In general, studies provided an adequate description of sample preparation, experimental analysis, and instrumental performance. No study provided an acceptable description of method validation. Three of the eight studies that analysed the relative quantification of metabolites were identified used internal standards [13,14,15], and five studies described methods used for assessing instrument variation [11,13,14,20,21]. Of the five studies that used absolute quantification, four did not report accuracy or precision validation data for their method on the instrument [9,17,19,22], while one study reported the limits of quantification and detection of the method [18]. Six studies provided a detailed description of data pre-processing [13,14,15,19,20,21]. Level one metabolite identification was reported in two studies [17,18]. Level two metabolite identification was reported in nine studies [9,11,13,14,15,16,19,20,23].

### 2.2. Urinary VOCs

A total of 48 cancer-associated urinary VOCs were reported within the 13 identified studies, with significant variation observed between different cancer types. VOCs belonged to 11 chemical classes (Figure 2). Five of the VOCs, 2,6-dimethyl-7-octen-2-ol, p-cresol, phenol, acetic acid, and dimethyl disulphide, were reported in more than one study as being associated with cancer (Table 3).

For prostate cancer, 29 urinary VOCs from nine chemical classes were identified (Figure 2 and Table 3). The majority of identified VOCs were aromatic compounds, ketones, and organic acids (Figure 2). Most VOCs showed decreased concentrations in the urine of prostate cancer patients compared to the urine of non-cancer patients. Alcohols, ketones, and organic acids were reported to have the largest decrease in concentration in prostate cancer. In comparison, aldehydes and siloxanes were identified to be increased in the urine of prostate cancer patients (Figure 3).

For gastrointestinal cancers (gastroesophageal, colorectal, and hepato-biliary), 21 urinary VOCs from eight chemical classes were identified, 19 of which were not identified in prostate cancer urine (Figure 2 and Table 3). The majority of the gastrointestinal cancer-related VOCs were aromatic compounds, alcohols, and ketones (Figure 2). Enol ether and organosulfur compounds were unique to the urine of gastrointestinal cancer patients. Compare to prostate cancer, the majority of the gastrointestinal cancer-related VOCs showed increased concentrations, particularly aromatic VOCs. Alcohols, ketones and organosulfur compounds were identified to be decreased in the urine of gastrointestinal cancer patients (Figure 3).

Six leukaemia and lymphoma-related urinary VOC biomarkers were reported by a single study [23]. These VOCs were identified to be increased in the urine of patients with these haematological malignancies, except for anisole, which was decreased in the urine of lymphoma patients. Formaldehyde were reported to be associated with bladder cancer by a single study [17]. No urinary VOC biomarker for lung cancer was reported. (Table 3).

### 2.3. Metabolic Analysis

Metabolic pathway analysis showed different levels of VOCs related to KEGG metabolic pathways that belonged to carbohydrate, lipid, amino acid, and energy metabolism. Glycolysis and the gluconeogenesis pathway had the most significant impact among metabolic pathways related to cancer specific urinary VOCs. Prostate cancer was associated with seven metabolic pathways categorised into carbohydrate, lipid, and amino acid metabolism. Gastrointestinal cancer was associated with six metabolic pathways belonging to the three pathway categories as well as the energy metabolism. Gastrointestinal cancer had a greater association with carbohydrate metabolism and energy metabolism compared to prostate cancer, suggesting different underlying metabolic profiles of these cancers. (Figure 4 and Appendix A).

## 3. Discussion

This systematic review provides an overview of the use of urinary VOCs for cancer diagnosis. The principal findings were a description of characteristic cancer associated urinary VOC biomarkers; promising diagnostic performance of urinary VOCs for the detection of prostate and gastrointestinal cancers; and a lack of standardisation in reported practices for urinary VOCs analysis.

Early detection is one of the most important factors influencing cancer survival. Many cancers, including those identified in this review, present with vague symptoms leading to a delay in their investigation and detection. Late diagnosis is associated with worse overall survival. For patients diagnosed with advanced (stage IV) prostate cancer, five-year survival is 49% compared to almost 100% for patients with early (stage I/II) disease [25]. Therefore, it is crucial to develop accurate, acceptable, and affordable methods for cancer early detection.

GC-MS was the most common analytical platform used for urinary VOC analysis, with multiple sampling techniques aiding pre-analysis VOC extraction. However, the majority of the studies failed to report adequate information concerning patient recruitment and study design, including strategies for mitigation of bias. No study performed an adequate validation of results. Three of the 13 studies validated their initial discoveries in independent cohorts [13,15,18]. Four studies applied internal standard normalisation to data [13,14,15,18], and two reported a use of quality control measures [13,14]. Therefore, future studies are needed to establish an “optimal” method for urinary VOC analysis. These studies should acknowledge the importance of standardisation and adoption of quality control measures to ensure accuracy and reproducibility.

The majority of studies identified by this review investigated urinary VOCs of prostate and gastrointestinal cancers. The wide variation in reported cancer-associated VOCs likely reflects diversity in the underlying tumour metabolic profiles and/or methodological variability secondary to the lack of standardised practices for urinary VOCs analysis. Protocols for the analysis of urinary metabolites have been published previously [26,27,28]. Whilst the current review was not intended to establish similar guidance, important considerations for the specific analysis of urinary VOCs are summarised in Table 4.

In general, both prostate and gastrointestinal cancer-specific VOCs had high sensitivity and specificity for cancer detection. As previously mentioned, a limited number of studies validated their findings in an independent dataset. In prostate cancer, the majority of cancer-specific VOCs were decreased compared to controls. In comparison, VOCs that were associated with gastrointestinal cancers tended to be found at elevated urinary levels. Identified VOCs originated from a wide variety of metabolic pathways, including carbohydrate metabolism, lipid metabolism, amino acid metabolism, and energy metabolism. Metabolic analysis derived provisional evidence that the metabolic pathways of both prostate and gastrointestinal cancer specific VOCs were different, with the latter being association more with carbohydrate metabolism and energy metabolism.

The origin of VOCs within the body and the mechanism by which they enter the urine remains incompletely understood. It is presumed that cancer-specific VOCs are of endogenous origin and produced as a result of abnormal metabolism either within the tumour itself or related tissues. VOCs released by tissues may travel in the systemic circulation from where they may be excreted via the lungs, skin, or renal tract. In the case of prostate cancer, there may also be local release directly from the prostate gland into urine. It has been hypothesised that the tumour-associated intestinal microbiome may also contribute to VOC production in gastrointestinal cancers. Further studies are needed to determine the underlying mechanisms of VOC production in cancer and the kinetics of their release.

This systematic review suffers from a number of limitations that principally concern the relatively small number of published studies within this field. A wide variation in the methodologies used by individual studies was observed, making it difficult to draw strong conclusions. Studies rarely utilised robust quality control strategies, and few studies validate findings within an independent patient cohort. Inadequate reporting of clinical parameters, including cancer stage, made it difficult to evaluate the performance of urinary VOC analysis on diagnosing early-stage cancer. For those studies that did report cancer stage, it was evident that the majority of enrolled patients had locally advanced disease. Therefore, observed metabolic differences can not been seen to truly represent “early” disease that is the ultimate target of the test. It should also be noted that the majority of the articles in this review originated from Europe and may therefore not be representative of other populations.

This systematic review summarises the progress of urinary VOC analysis for the diagnosis of cancer. Although there were variations in study quality, urinary VOC analysis exhibited promising performance for developing non-invasive diagnostic tools for cancer diagnosis and demonstrating metabolic profiling of different types of cancer. In order to develop future studies and translate their results to large clinical practice, the methodological weakness and limitations summarised in this review must be overcome.

## 4. Materials and Methods

### 4.1. Literature Search

A systematic online literature search was conducted to identify all studies that measured differences in urinary VOCs between cancer patients and relevant controls in accordance with the recommendations of the Cochrane library and Meta-analysis of Observational Studies in Epidemiology (MOOSE) guidelines [29]. Databases that were searched included Medline (1946–13th December 2019) via OvidSP, Ovid Embase (1947–13th December 2019), and Cochrane Library. The following terms were used in the search strategy: urine, volatile organic compounds, biomarkers, metabolomics, metabolic profiling, magnetic resonance spectroscopy; mass spectrometry, and carcinoma. All variations in spelling including truncated search term using wild card characters and the “related articles” function were used in combination with the Boolean operators AND and OR. Full information of search strategy is provided as an online Appendix A. Reference lists of qualified articles were screened to include potentially relevant studies.

Two independent reviewers, Q.W. and P.B., screened the titles and abstracts of all studies identified through database searching. The full text of potentially relevant articles was reviewed for eligibility. Only original research articles published in the English language were considered. Included studies were those that identified potential VOC biomarkers of cancers by profiling the urine of patients and relevant controls (patients without cancer) using mass spectrometry based technologies. Studies were excluded if they did not report named VOC biomarkers or if they reported results from mixed cancer cohorts where the results of each subtype could not be clearly separated. Review articles, conference abstracts, articles not written in the English language, and animal and cell studies were excluded. A third reviewer (G.B.H.) was consulted when disagreement in study inclusion arose.

### 4.2. Outcome Measures

The following information from included articles was extracted and summarised: year of publication, country of origin, study design, recruitment period, number of participants, cancer type, analytical platform used, sampling technique methodology, quality control method(s), normalisation method(s), number of VOCs identified, identity of VOCs increased/decreased in cancer, method of statistical analysis (including prediction model used), sensitivity and specificity, and area under the receiver operating characteristic (AU-ROC) curve.

### 4.3. Quality Assessment

Quality of all the studies was assessed using 3 tools. The Quality Assessment of Diagnostic Accuracy Studies-2 (QUADAS-2) tool was used to assess the risk of bias and applicability of the study [30]. The QUADAS-2 was divided into risk of bias of patient selection, diagnostic test, reference standard and patient flow and timing. This test also investigated the applicability of patient selection, diagnostic test, and reference standard to the systematic review. The Standards for Reporting of Diagnostic Accuracy studies (STARD) tool was used to assess reporting quality of the study [31]. This tool evaluated all the sections including title, abstract, introduction, methods, results and discussion to provide a comprehensive figure of the completeness and transparency of reporting of diagnostic accuracy studies. The Chemical Analysis Working Group (CAWG)-Metabolomics Standard Initiative (MSI) criteria were used to assess the quality of metadata of the study [24]. CAWG-MSI proposed minimum reporting standards related to the chemical analysis aspects of metabolomics experiments including sample preparation, experimental analysis, quality control, metabolite identification, and data pre-processing.

### 4.4. Metabolic Analysis

All VOCs identified were checked and classed according to the Kyoto Encyclopaedia of Genes and Genomes (KEGG) pathway and the Human Metabolome Database (HMDB). Statistical analysis was performed using the pathway analysis module in MetaboAnalyst 4.0, which is based on the R programming language (version 3.5.3, The R Project for Statistical Computing, www.r-project.org). VOCs that were significantly increased or decreased in each study were selected. In the pathway analysis module, compound names were standardised against HMDB, KEGG, and PubChem to match well-annotated compounds in KEGG pathway libraries. Based on KEGG pathway libraries, parameters used to analyse data were a hypergeometric test for over representation analysis and a relative-betweeness centrality test for pathway topology analysis [32,33,34]. Normalisation was performed following an equation for weighted means of each identified VOC: the proportion of the total number of VOCs identified per study, divided by the total number of VOCs identified in each cancer type, then multiplied by the total number of studies in which this VOC was identified (Appendix A).

## Figures and Tables

**Figure 1 metabolites-11-00017-f001:**
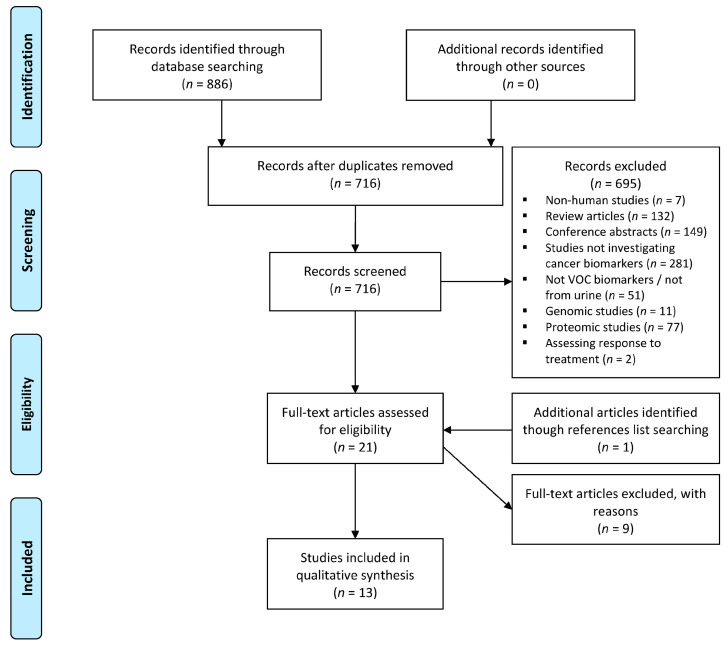
Preferred Reporting Items for Systematic Reviews and Meta-Analyses (PRISMA) flow diagram. VOC: volatile organic compound. The systematic literature search identified 886 studies, with 13 studies included after screening and assessment for eligibility.

**Figure 2 metabolites-11-00017-f002:**
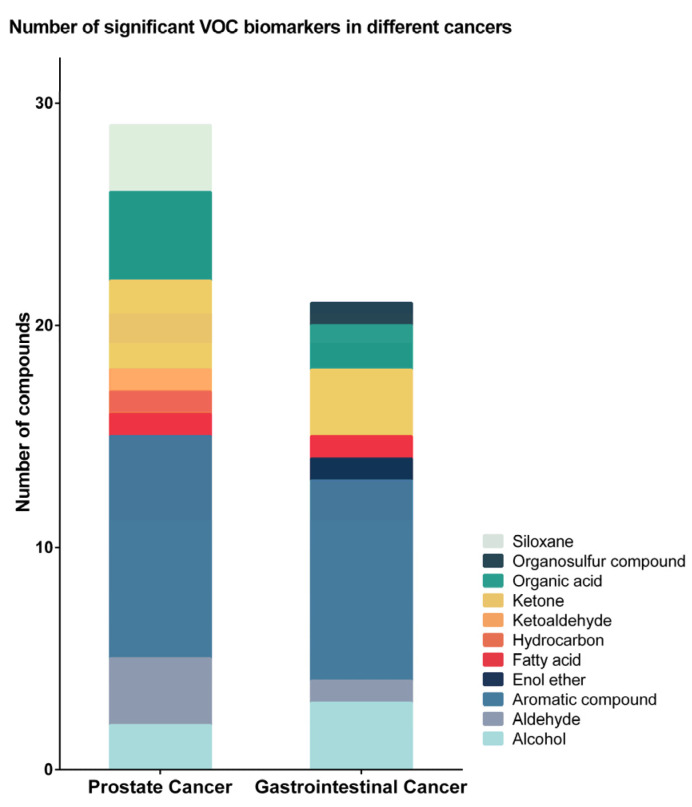
Number of identified volatile organic compounds (VOCs) in different cancers. For prostate cancer, 29 urinary VOCs from nine chemical classes were identified, and the majority of them were aromatic compounds, ketones, and organic acids. For gastrointestinal cancer, 21 VOCs from eight chemical classes were identified, 19 of which were not identified in prostate cancer urine. The majority were aromatic compounds, alcohols, and ketones.

**Figure 3 metabolites-11-00017-f003:**
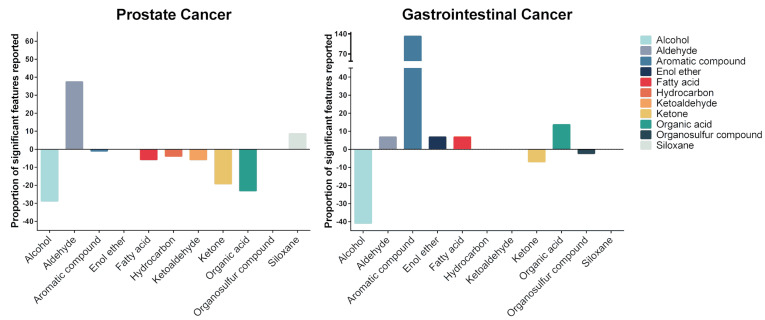
Proportion of identified compound chemical classes in different cancers. Most prostate cancer-related VOCs showed decreased concentrations in the urine of prostate cancer patients compared to the urine of non-cancer patients. The majority of the gastrointestinal cancer-related VOCs showed increased concentrations, particularly of aromatic VOCs.

**Figure 4 metabolites-11-00017-f004:**
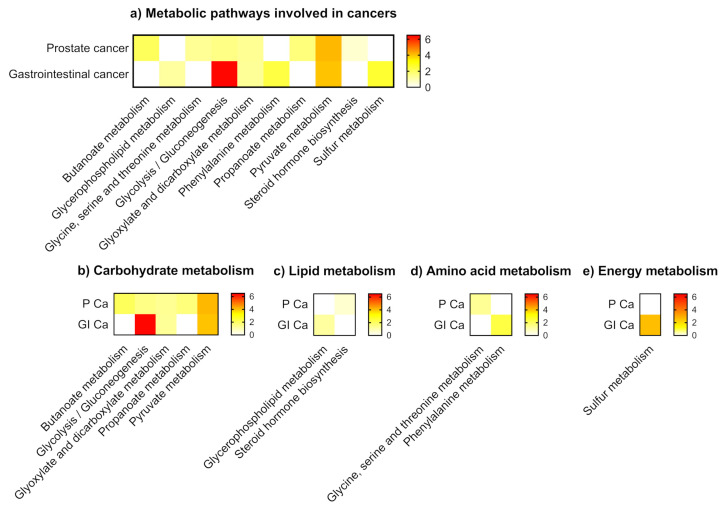
Metabolic pathways involved in cancers. (**a**) All metabolic pathways (**b**) Carbohydrate metabolism (**c**) Lipid metabolism (**d**) Amino acid metabolism (**e**) Energy metabolism. NOTE: P Ca: prostate cancer; GI Ca: gastrointestinal cancer. Gastrointestinal cancer had a greater association with carbohydrate metabolism and energy metabolism compared to prostate cancer, suggesting different underlying metabolic profiles of these cancers.

**Table 1 metabolites-11-00017-t001:** Study characteristics. Urinary volatile organic compound analysis for cancer diagnosis: analytical and biostatistical techniques for biomarker discovery.

Author	Year	Country	Cancer Type	Cancer Stage	No. Patients (cancer/all)	Sample Type	Analytical Platform	Sampling Technique	Prediction Model	Sensitivity (%)	Specificity (%)	AUC	Ref.
**Lima**	2019	Portugal	Prostate	I, II, IIa, IIb, III, IV	58/118	Headspace	GC-MS	SPME, PFBHA derivatisation	PLS-DA, ROC	72 (Training) 89 (Validation)	96 (Training) 83 (Validation)	0.856 (Training) 0.904 (Validation)	[13]
**Struck-Lewicka**	2014	Poland	Prostate	Not reported	32/64	Liquid phase	HPLC-ESI-TOF-MS; GC-MS	LC: Centrifugation; GC: BSTFA derivatisation	PCA, PLS-DA	-	-	-	[14]
**Gao**	2019	USA	Prostate	Not reported	108/183	Liquid phase	TD-GC-MS	Stir bar sorptive extraction	ROC	96 (Training), 87 (Validation)	80 (Training), 77 (Validation)	0.92 (Training), 0.86 (Validation)	[15]
**Jimenez-Pacheco**	2018	Spain	Prostate	Not reported	29/50	Headspace	GC-MS	Dynamic headspace SPME	-	-	-	-	[16]
**Khalid**	2015	UK	Prostate	Not reported	59/102	Headspace	GC-MS	SPME	Repeated 10-Fold CV, Repeated Double CV	-	-	-	[11]
**Spanel**	1999	Czech Republic	Prostate, Bladder	Not reported	38/52	Headspace	SIFT-MS	Direct sampling	-	-	-	-	[17]
**Chen**	2016	China	GI (G)	Ia/b, IIa/b, IIIa/b/c, IV	159/293	Liquid phase	GC-MS	Derivatisation	OPLS-DA	77.4 (Validation)	85.1 (Validation)	0.893 (Validation)	[18]
**Navaneethan**	2015	USA	GI (HPB)	Not reported	15/54	Liquid phase	SIFT-MS	Centrifugation	Logistic regression model, ROC	93.3	61.5	0.83	[19]
**Panebianco**	2017	Italy	GI (G, C, HPB)	Not reported	23/38	Headspace	GC-TOF-MS; GC-qMS; GC/O	SPME	-	-	-	-	[20]
**Arasaradnam**	2014	UK	GI (CR)	Not reported	83/133	Headspace	FAIMS; GC-MS	Direct sampling; automated pre-concentration	Fisher discriminant analysis	88	60	-	[21]
**Huang**	2013	UK	GI (O, G)	Not reported	17/44	Headspace	SIFT-MS	Direct sampling	ROC	-	-	0.904	[9]
**Rozhentsov**	2014	Russia	Lung GI (O, G, C)	Not reported	46/81	Headspace	GC-MS	Headspace SPME (three phase micro-extraction (TPME))	Image analysis	100	90.62	-	[22]
**Silva**	2011	Portugal	Leukaemia; Lymphoma; GI (CR)	Not reported	33/54	Headspace	GC-qMS	dHS-SPME	PCA	-	-	-	[23]

GI: gastrointestinal cancer; O: oesophageal cancer; G: gastric cancer; HPB: Hepato-biliary cancer; C: colon cancer; CR: colorectal cancer; GC-MS: gas chromatography mass spectrometry; TD: thermal desorption; HPLC-ESI-TOF-MS: high-performance liquid chromatography electrospray ionisation time of flight mass spectrometry; TOF: time of flight; SIFT-MS: selected ion flow tube mass spectrometry; GC/O: gas chromatography-olfactometry; FAIMS: field asymmetric ion mobility spectrometry; qMS: quadrupole mass spectrometry; SPME: solid phase microextraction; PFBHA: O-(2,3,4,5,6-pentafluorobenzyl)hydroxylamine; BSTFA: N,O-Bis(trimethylsilyl)trifluoroacetamide; dHS-SPME: dynamic solid-phase microextraction in headspace mode; PCA: principal component analysis; PLS-DA: partial least squares-discriminant analysis; OPLS-DA: orthogonal partial least squares discriminant analysis; ROC: receiver operating characteristic; Repeated 10-Fold CV: repeated 10-fold cross-validation; Repeated Double CV: repeated double cross-validation.

**Table 2 metabolites-11-00017-t002:** Quality assessment with Standards for Reporting of Diagnostic Accuracy studies (STARD), Quality Assessment of Diagnostic Accuracy Studies-2 (QUADAS-2), and Chemical Analysis Working Group Metabolomics Standard Initiative (CAWG-MSI) score. There was an overall low risk of bias and high applicability of the 13 studies to the review question. The completeness and transparency of reporting was inadequate. There was an overall inadequate reporting of metadata of the studies.

	Study	Overall Diagnostic Quality	QUADAS							STARD Score	CAWG-MSI Metadata
			Risk of Bias				Applicability Concerns				
			Patient Selection	Index Test	Reference Standard	Flow and Timing	Patient Selection	Index	Reference Standard		
Prostate cancer (PCa)	Lima 2019	Good	Low	Low	Low	Low	Low	Low	Low	26	23
	Struck-Lewicka 2014	Fair	Unclear	Unclear	Unclear	Low	Low	Low	Low	14	20
	Gao 2019	Good	Unclear	Unclear	Unclear	Low	Low	Low	Low	27	18
	Jimenez-Pacheco 2018	Good	Low	Unclear	Low	Low	Low	Low	Low	19	14
	Khalid 2015	Good	Low	Unclear	Low	Low	Low	Low	Low	23	15
PCa, bladder cancer	Spanel 1999	Fair	Low	Unclear	Unclear	Low	Low	Low	Unclear	12	9
GI cancer	Chen 2016	Good	Low	Unclear	Low	Low	Low	Low	Low	23	22
	Navaneethan 2015	Good	Low	Unclear	Low	Low	Low	Low	Low	23	25
	Panebianco 2017	Fair	Unclear	Unclear	Low	Low	Low	Low	Low	18	13
	Arasaradnam 2014	Fair	Unclear	Unclear	Low	Low	Low	Low	Low	21	15
	Huang 2013	Fair	Low	Low	Low	Low	Low	Low	Low	21	9
Lung, GI cancer	Rozhentsov 2014	Fair	Unclear	Unclear	Low	Low	Low	Low	Low	16	6
Leukaemia, colorectal cancer, lymphoma	Silva 2011	Fair	Unclear	Unclear	Low	Low	Low	Low	Low	16	15

**Table 3 metabolites-11-00017-t003:** List of all volatile organic compounds (VOCs), their chemical class, and studies that identified them to be increased or decreased in cancers.

Compound Name	Chemical Class	Study	Prostate Cancer		Gastrointestinal Cancer	Leukaemia	Lymphoma	Bladder Cancer
			Increased/Decreased in a Cancer					
2,6-dimethyl-7-octen-2-ol	Alcohol	Jimenez-Pacheco 2018; Khalid 2015	↓					
2-propanol		Navaneethan 2015			↓			
Ethanol		Navaneethan 2015			↓			
Methanol		Huang 2013			↑			
2-ethylhexanol		Jimenez-Pacheco 2018	↓					
Formaldehyde	Aldehyde	Spanel 1999	↑					↑
Acetaldehyde		Huang 2013			↑			
Hexanal		Lima 2019	↓					
Pentanal		Khalid 2015	↑					
1-(2,4-Dimethylphenyl)-3-(tetrahydrofuryl-2)propane	Aromatic compound	Gao 2019	↓					
1,2-dihydro-1,1,6-trimethyl-naphthalene		Silva 2011			↑	↑	↑	
Dihydroedulan IA		Lima 2019	↓					
3-Phenylpropionaldehyde		Lima 2019	↑					
Phenylacetic acid		Panebianco 2017			↑			
2,5-Dimethylbenzaldehyde		Lima 2019	↑					
3,5-dimethylbenzaldehyde		Jimenez-Pacheco 2018	↓					
p-xylene		Jimenez-Pacheco 2018	↑					
3-methylphenol (m-Cresol)		Jimenez-Pacheco 2018	↑					
p-cresol		Chen 2016; Silva 2011			↑	↑	↑	
Phenol		Jimenez-Pacheco 2018; Huang 2013	↑		↓			
4-ethyl guaiacol		Panebianco 2017			↓			
Anisole		Silva 2011			↑	↑	↓	
Furan		Jimenez-Pacheco 2018	↑	↓				
Thiophene		Panebianco 2017			↓			
p-cymene		Silva 2011			↑	↑	↑	
Indole		Struck-Lewicka 2014	↓					
2-methyl3-phenyl-2-propenal		Silva 2011			↑	↑	↑	
2-methoxythiophene	Enol ether	Panebianco 2017			↑			
Hexanoic acid	Fatty acid	Huang 2013			↑			
Butyric acid		Struck-Lewicka 2014	↓					
Santolina triene	Hydrocarbon	Jimenez-Pacheco 2018	↓					
Methylglyoxal	Ketoaldehyde	Lima 2019	↓					
2-butanone	Ketone	Jimenez-Pacheco 2018	↑					
2-octanone		Khalid 2015	↓					
3-methyl-2-pentanone		Panebianco 2017			↓			
3-octanone		Khalid 2015	↓					
4-(or 5-)methyl-3-hexanone		Panebianco 2017			↓			
4-Methylhexan-3-one		Lima 2019	↓					
Acetone		Huang 2013			↑			
Acetic acid	Organic acid	Struck-Lewicka 2014; Huang 2013	↓		↑			
Propenoic acid		Struck-Lewicka 2014	↓					
Isobutyric acid		Struck-Lewicka 2014	↓					
Propionic acid		Struck-Lewicka 2014	↓					
Hydrogen sulfide		Huang 2013			↑			
Dimethyl disulphide	Organosulfur compound	Silva 2011; Panebianco 2017			↓	↓	↓	
1,1,1,5,5,5-hexamethyl-3,3-bis[(trimethylsilyl)oxy]-Trisiloxane	Siloxane	Gao 2019	↑					
1,1,3,3,5,5,7,7,9,9-decamethyl-pentasiloxane		Gao 2019	↑					
Ethyl à-hydroxymyristate trisiloxane		Gao 2019	↓					

In Jimenez-Pacheco A. 2018, Furan is increased before prostate massage and is decreased after prostate massage.

**Table 4 metabolites-11-00017-t004:** Considerations for analysis of urinary volatile organic compounds.

Workflow	Analytical Step	Considerations	Ref.
Experimental design		Patient selection	
		Testing and independent validation cohort setup	[13,15,18]
Sample preparation	Urine collection and storage	Mode of collection (e.g., spot collection or 24h collection)	
		Choice of receptacle (e.g., appropriate volume, ultra-low-temperature friendly, no unwanted contaminants)	
		Sources of contamination	
		Sample filtration	
		Sample handling, aliquot, transfer and storage	
		Impact of freeze–thaw cycles	
	Sampling technique	Analytical phase (e.g., headspace, liquid phase)	
		Selection of sample extraction techniques (e.g., SPME, HiSorb, direct injection)	[9,11,13,15,16,17,20,21,22,23]
		Sample extraction optimisation (e.g., pH, dilution, salting out, temperature, agitation)	
		Sample preparation (e.g., derivatisation)	[13,14,18,19]
	Internal standards and QC samples	Appropriate internal standards	[13,14,15,18]
		QCs (pooled sample QCs, synthetic urine, spiked urine. Monitor and correction of analytical variability)	[13,14]
MS analysis	Analytical platform	Selection of analytical platform (e.g., GC-MS, SIFT-MS, LC-MS)	
		Selection of column	
		Automatic or manual injection	
		Optimisation of separation parameters (e.g., column dimensions, gradients, temperatures, flow rates, etc.)	
		Selection of ionisation (e.g., EI, ESI) and mass analyser (high mass resolution MS (TOF, qTOF), low mass resolution MS (single and triple quadrupoles and quadrupole ion-traps))	
		Optimisation of MS parameters (e.g., m/z range, mass resolution)	
	MS data collection	Run order (e.g., use of randomised block design)	
Data analysis	Data preprocessing	Peak alignment	
		Peak detection, integration and identification	
		Removal of irreproducible, non-linear, and contaminant compounds	
	Statistical analysis	Descriptive statistics used	
		Univariate analysis used	[9,11,13,14,15,16,18,19,20,23]
		Multivariate analysis used (e.g., PCA, PLS-DA, OPLS-DA)	[11,13,14,15,18,19,21,23]
		Prediction model used (e.g., ROC analysis)	[9,13,15,18,19]

SPME: solid phase microextraction; QC: quality control; GC-MS: gas chromatography mass spectrometry; SIFT-MS: selected ion flow tube mass spectrometry; LC-MS: liquid chromatography mass spectrometry; EI: electron ionisation; ESI: electrospray ionisation; TOF: time of flight; PCA: principal component analysis; PLS-DA: partial least squares-discriminant analysis; OPLS-DA: orthogonal partial least squares discriminant analysis; ROC: receiver operating characteristic.

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
