# Peer review of "Urinary Volatile Organic Compound Analysis for the Diagnosis of Cancer: A Systematic Literature Review and Quality Assessment"

_metabolites, 2020, doi:10.3390/metabo11010017_

Round 1

Reviewer 1 Report

This review summarises the published literature concerning urinary VOCs associated with cancer, including VOC biomarkers of five cancer subtypes, analytical platforms used for urinary VOC analysis, the origin of VOCs within the body and the mechanism by which they enter the urine.

This overview is quite critical, as is required when the wide variation in reported cancer-associated VOCs most likely reflect the diversity in the underlying tumour metabolic profiles and/or methodological variability, together with a lack of standardisation in reported practices. Only three of the evaluated studies validated their initial discoveries in independent cohorts.

Nevertheless, this review will be of interest to the research community, since the topic of urinary VOCs in relation to disease is currently attracting interest. I recommend the review for publication in METABOLITES.

Author Response

We are grateful to the reviewer for providing thoughtful review and insightful comments. We hope our input to the topic of cancer-related urinary VOCs can help research community in developing future studies and translating their results to large clinical practice.

Reviewer 2 Report

This current paper is a review of urinary volatile organic compound analysis for the diagnosis of cancer: a systematic literature review and quality assessment. This systematic review is informative not only for the methodology of VOC detection, but also indicated that urine VOCs can serve as potential cancer biomarkers. However, this reviewer has a few concerns on this review context, and there are multiple points that need to be clarified.

  • In this review, three types of tools were used for assessing quality in 2.3. Quality assessment and Table S1. However, the introduction of each method and the selected parameter is needed to be elaborated in 2.3.
  • Several studies in Table 1 had prediction models to identify urinary VOCs as cancer biomarkers. Authors may consider including more information such as the types/classes of VOCs, stages of cancer, and/or any treatment types.
  • There is an over-reliance on supplemental data such as Table S1. The supplemental data is frequently referenced throughout the manuscript, and 3.2. Urinary VOCs for discussion of potential biomarkers. If the supplemental data is this critical, it should not be supplemental.
  • Title and/or caption of Figures and Tables should contain briefly description of the summarized results.
  • The contents in Table 2 are too general. Used parameters in each reference should be more elaborated in Table 2 or in the manuscript, and the pros and cons of choosing these parameters should be briefly discussed.

Minor concerns

Line 11: Correct to: “cancer-associated”

Line 18: Correct to: “The metabolic analysis”

Line 22: Correct to: “have”

Line 38: Correct to: “samples”

Line 56: Correct to: “the methodological”

Line 78: Correct to: “excluded”

Line 81: Correct to: “was”

Line 89: Correct to: “the risk”

Line 90: Correct to: “Study”

Line 233: Correct to: “to determine”

Author Response

We thank to the reviewer for providing thoughtful comments and efforts towards
improving our manuscript. Please see the attachment for our reply to the comments.
